# METACOGNITIVE REUSE: TURNING RECURRING LLM REASONING INTO CONCISE BEHAVIORS

## ABSTRACT

Large language models (LLMs) now solve multi-step problems by emitting extended chains of thought. During the process, they often re-derive the same intermediate steps across problems, inflating token usage and latency. This saturation of the context window leaves less capacity for exploration. We study a simple mechanism that converts recurring reasoning fragments into concise, reusable "behaviors" (name + instruction) via the model's own *metacognitive analysis of prior traces*. These behaviors are stored in a "behavior handbook" which supplies them to the model in-context at inference or distills them into parameters via supervised fine-tuning. This approach achieves improved test-time reasoning across three different settings - 1) **Behavior-conditioned inference**: Providing the LLM relevant behaviors in-context during reasoning reduces number of reasoning tokens by up to 46% while matching or improving baseline accuracy; 2) **Behavior-guided self-improvement**: Without any parameter updates, the model improves its own future reasoning by leveraging behaviors from its own past problem solving attempts. This yields up to 10% higher accuracy than a naive critique-and-revise baseline; and 3) **Behavior-conditioned SFT**: SFT on behavior-conditioned reasoning traces is more effective at converting non-reasoning models into reasoning models as compared to vanilla SFT. Together, these results indicate that turning slow derivations into fast procedural hints enables LLMs to remember how to reason, not just what to conclude.

## 1 INTRODUCTION

LLMs have made rapid progress on mathematics, coding and other multi-step tasks by generating long, deliberative chains-of-thought (Wei et al., 2022; Guo et al., 2025; Shao et al., 2024; OpenAI, 2024; Muennighoff et al., 2025; Ye et al., 2025; Gao et al., 2024; Lambert et al., 2024; Team et al., 2025). Yet, this capability exposes a structural inefficiency: each new problem triggers reconstruction of ubiquitous sub-procedures (e.g., finite-series sums, case splits, unit conversions), inflating token usage and latency. For instance, suppose the LLM derives the finite geometric series formula while solving one problem. Can it avoid re-deriving from scratch when similar reasoning is needed for another problem? Current inference loops lack a mechanism to promote frequently rediscovered patterns into a compact, retrievable form.

We introduce a *metacognitive pathway* that extracts and reuses such patterns. Given a problem, the model first solves it, then reflects on its trace to identify generalizable steps, and finally emits a set of behaviors—short, actionable instructions with canonical names. These behaviors populate a searchable handbook (a procedural memory) that can be provided in-context at test time or internalized through supervised fine-tuning. This provides a framework for turning verbose derivations into *quick reflexes*.

Unlike typical memory/Retrieval-Augmented Generation (RAG) systems that store declarative facts, the handbook targets procedural knowledge (Willingham et al., 1989) about how to think. This procedural memory contrasts sharply with most existing "memory" add-ons for LLMs, including RAG, which target declarative knowledge for tasks such as factual question-answering (Borgeaud et al., 2022; Lewis et al., 2020). Instead of being assembled from curated documents or knowledge graphs—rather, it is generated by the model itself. It emerges from the model's own metacognitive cycle: critiquing its own chain-of-thought and abstracting repeated reasoning patterns into behaviors.

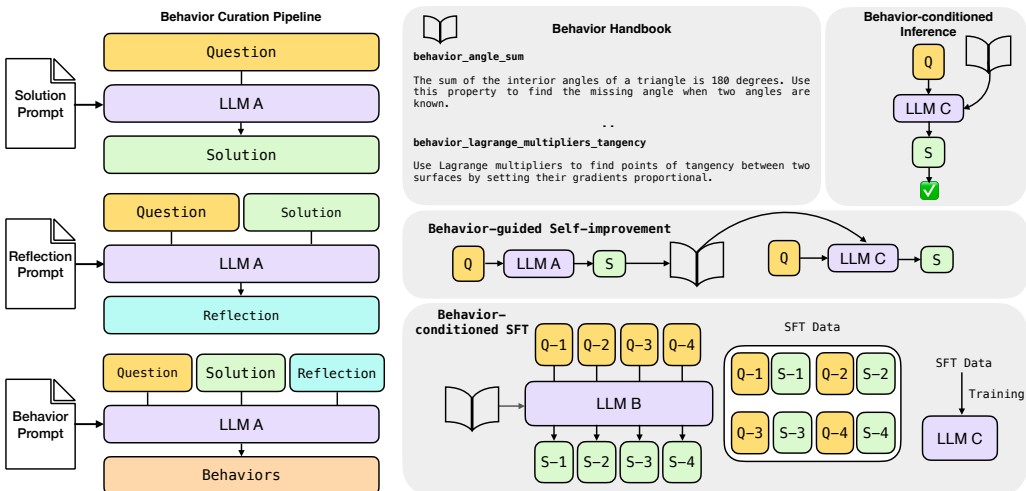

Figure 1: **Behavior Curation Pipeline (left)**: All the 3 stages of behavior curation pipeline are shown. The stages are described in detail in Section 3 with detailed prompts for each stage shown in Figure A.1. LLM A refers to the `Metacognitive Strategist`. **Reasoning with behaviors (right)**: This part showcases various ways in which behaviors are utilized for LLM reasoning. For behavior-conditioned inference and behavior-guided self-improvement, behaviors retrieved from the behavior handbook, are provided in-context to the `Student` model (LLM C) during inference. For behavior-conditioned SFT, a training dataset is created using a `Teacher` LLM (LLM B) via behavior-conditioned inference and then a `Student` LLM is trained on the resulting (question, response) pairs. After training, during inference, the `Student` LLM is queried with the given question without any behaviors in-context.

We evaluate three instantiations of the proposed framework. (i) Behavior-conditioned inference: Providing behaviors obtained by solving questions in-context results in reasoning chains that utilize up to 46% fewer tokens while improving or maintaining strong performance across MATH and AIME benchmarks (ii) Behavior-guided self-improvement: While solving a problem, providing the model access to behaviors extracted by itself from its own past attempts for that question improves accuracy by up to 10% compared to naive self-improvement baseline. (iii) Behavior-conditioned SFT: training on reasoning traces generated via behavior-conditioned inference yields models that are both more accurate and more concise than models trained on ordinary traces, especially when turning non-reasoning models into reasoning models.

Contributions.

1. We formalize behaviors as named, reusable reasoning instructions discovered by metacognitive reflection over solution traces.

2. We introduce a three-step approach to employ an LLM to extract behaviors from its own reasoning traces (Section 3).

3. We develop three settings for utilizing these behaviors: behavior-conditioned inference, behavior-guided self-improvement, and behavior-conditioned SFT. (Section 4)

4. We provide empirical evidence of the effectiveness of our behavior-based approach across each of the three settings, evaluated on challenging mathematical benchmarks such as MATH (Hendrycks et al., 2021) and AIME–24/25 (MAA, 2024; 2025) (Section 5).

5. We discuss some limitations and challenges of the proposed framework—e.g., lack of dynamic behavior retrieval during long solutions, usage of behaviors beyond math, constructing large scale behavior handbooks across various domains, etc.

By converting frequently rediscovered steps into compact procedures, behavior handbooks encourage LLMs to remember how to think. This simple addition to the reasoning stack improves token efficiency and suggests a path toward models that accumulate procedural knowledge over time.

## 1.1 PAPER OUTLINE

The paper starts with describing the pipeline for curating the behaviors (Section 3) followed by various ways in which behaviors are utilized for improved reasoning (Section 4). The experiment section describes the corresponding experiment results. Section 5.1 presents results for behavior-conditioned inference on the MATH (Hendrycks et al., 2021) and the AIME datasets (MAA, 2024; 2025) showing that the proposed approach exhibits similar or improved performance compared to normal inference while reducing token usage by up to 46%. Section 5.2 presents the self-improvement experiment where the behavior-guided approach uses behaviors as *lessons* for scalable self-improvement achieving up to 10% higher accuracy as compared to the baseline self-improvement approach at the highest considered token budget of 16,384. Finally, the SFT experiments (Section 5.3) show that using behavior-conditioned inference to generate reasoning traces for SFT results in stronger reasoning models as compared to performing SFT with vanilla reasoning traces.

## 2 RELATED WORK

**Efficient Reasoning with LLMs** Reinforcement-tuned, long-form chain-of-thought (CoT) prompting has enabled LLMs to solve complex tasks in math, logic, and code (OpenAI, 2024; Guo et al., 2025; Shao et al., 2024; Gao et al., 2024; Lambert et al., 2024; Team et al., 2025; Zeng et al., 2025a; Song et al., 2025; Muennighoff et al., 2025; Ye et al., 2025). Recent work aims to shorten these traces without losing accuracy, e.g., Skeleton-of-Thought (outline then expand) (Ning et al., 2023), TokenSkip (drop redundant tokens) (Xia et al., 2025), Dynasor (early-exit probes) (Fu et al., 2024), and MinD (concise single-trajectory blocks) (Zeng et al., 2025b). Our approach shares this efficiency goal but differs in two ways: (i) efficiency emerges naturally as the model abstracts recurring reasoning into reusable behaviors, and (ii) these behaviors also improve accuracy, as shown in Section 5.3, where behavior-conditioned traces outperform long-form CoT training.

**Metacognitive abilities of LLMs** Metacognition refers to humans' "thinking about thinking" (Flavell, 1979). Didolkar et al. (2024) suggested that one LLM analog of metacognition is the ability to extract reusable "skills" from the CoT of LLMs, and showed that frontier LLMs can extract meaningful skill catalogs from task datasets. Such LLM-extracted skill catalogs were used to create more difficult math questions in Shah et al. (2025), by requiring questions to involve skill compositions. Kaur et al. (2025) follow a similar approach for instruction-following skills. He et al. (2025) use skill categories to study in-context learning in smaller language models. The novelty of our work is to apply metacognitive thinking to help reasoning models with their longer and complicated reasoning traces.

**Memory in LLMs** Current memory implementations for LLM mainly rely on an external store of factual knowledge (such as Wikipedia) that the model can search (Borgeaud et al., 2022; Lewis et al., 2020; Guu et al., 2020; Shi et al., 2023; He et al., 2021). Retrieval-augmented generation pulls passages from this factual memory at inference time and conditions the decoder on the evidence to answer knowledge-intensive queries. More recently, retrieval has been woven directly into multi-step reasoning, with methods like ReAct and IR-CoT interleaving "think→retrieve→think" loops to blueuce hallucinations (Yao et al., 2023; Trivedi et al., 2022). These implementations of memory mainly store declarative knowledge which corresponds to *what* is true, not *how to* think. Procedural knowledge—skills and routines acquiblue through repetition—remains largely unexploblue. Our proposed behavior handbook is one instantiation of such procedural memory for LLMs: it captures how-to strategies distilled from repeated reasoning patterns and stores them for future reuse.

## 3 BEHAVIORS FROM REASONING TRACES

Reasoning LLMs emit a long chain-of-thought (CoT) which we will also refer to as a reasoning trace. We define a *behavior* as a reusable skill—a concise piece of knowledge—distilled from an LLM's chain of thought. Such behaviors can be invoked across tasks to make inference-time reasoning both faster and more accurate. Each behavior is represented as a (`name, instruction`) pair. For example:

systematic_counting → Systematically count possibilities by examining each digit's contribution without overlap; this prevents missed cases and double-counts.

The remainder of this section describes the process of deriving behaviors from LLM-generated reasoning traces.

Figure 1 depicts the entire pipeline. The framework employs LLMs in 3 different roles - 1) A Metacognitive Strategist (LLM A) which extracts behaviors from its own reasoning traces; 2) A Teacher (LLM B) which generates data for SFT training; and 3) A Student (LLM C) whose reasoning is aided by behaviors either via behavior-conditioned inference or behavior-conditioned SFT. We dive deeper into each of these roles in the following sections. First, we describe the working of the Metacognitive Strategist.

**Extracting Behaviors**  To extract behaviors, the Metacognitive Strategist produces a solution for a given question which consists of the reasoning trace + the final answer. The prompt for this interaction is shown in Figure A.1 (Solution Prompt). The question–solution pair is then fed to the Metacognitive Strategist again to generate a *reflection* which evaluates whether the reasoning is logically sound, the answer correct, and whether any new, reusable behaviors can be distilled to streamline future problem solving (See Reflection Prompt in Figure A.1). Finally, via another query the Metacognitive Strategist converts the question, solution, and reflection into a set of (name, instruction) behavior entries, which are appended to an ever-growing behavior handbook (See Behavior Prompt in Figure A.1). The behavior handbook panel in Figure 1 shows two example behaviors derived from the MATH (Hendrycks et al., 2021) and AIME–24 datasets (MAA, 2024) datasets.

The DeepSeek-R1-Distill-Llama-70B (R1-Llama-70B) (Guo et al., 2025) is used as the Metacognitive Strategist.

## 4 BEHAVIOR-GUIDED REASONING

This section discusses various ways in which the behavior handbook is utilized for scalable and efficient reasoning.

### 4.1 BEHAVIOR-CONDITIONED INFERENCE

One simple way to use behaviors is to provide a Student LLM with access to them in-context during reasoning (Figure 1), a process we call behavior-conditioned inference (BCI). For a question $Q_i$, the system retrieves relevant behaviors $B_i$ from the behavior handbook, then feeds these behaviors, their instructions, and the question into the LLM to produce a solution: $(B_i, Q_i) \rightarrow S_i$.

The BCI prompt format is shown in Figure A.2. The retrieval function varies by dataset. On MATH (Hendrycks et al., 2021), questions are annotated with 7 topics, so behaviors can be grouped by topic and retrieved accordingly. For datasets without topic labels (e.g., AIME-24/25 (MAA, 2024; 2025)), we use embedding-based retrieval: question and behaviors are embedded, ranked by cosine similarity, and the top-$K$ are selected. Details of each retrieval setup are given in Section 5.

### 4.2 BEHAVIOR-GUIDED SELF-IMPROVEMENT

Self-improvement is a given model's ability to improve its own reasoning. To achieve this, behaviors curated by a model from the reasoning traces of a particular question are then fed back into the model in-context to serve as lessons or hints to solve the same question or new questions. The implementation closely follows that of BCI and uses the same prompt. This process is depicted in Figure A.2. The Student LLM is the same as the Metacognitive Strategist.

### 4.3 BEHAVIOR-CONDITIONED SUPERVISED FINE-TUNING

Behavior-conditioned inference requires retrieval and extra prompt tokens at test time. To remove this overhead, we internalize behaviors via fine-tuning, a process we call Behavior-conditioned

supervised fine-tuning (BC-SFT). In this setup, the `Metacognitive Strategist` extracts behaviors, the `Teacher` generates behavior-conditioned responses using BCI, and the `Student` is fine-tuned on these (question, response) pairs.

As shown in Figure 1 (see Section 5.3 for details), the fine-tuned `Student` no longer needs explicit behavior prompts—it naturally applies the learned behaviors. This distillation turns the teacher's explicit, annotated reasoning into the student's faster, more intuitive, low-token solutions, enabling even non-reasoning models to acquire reasoning skills.

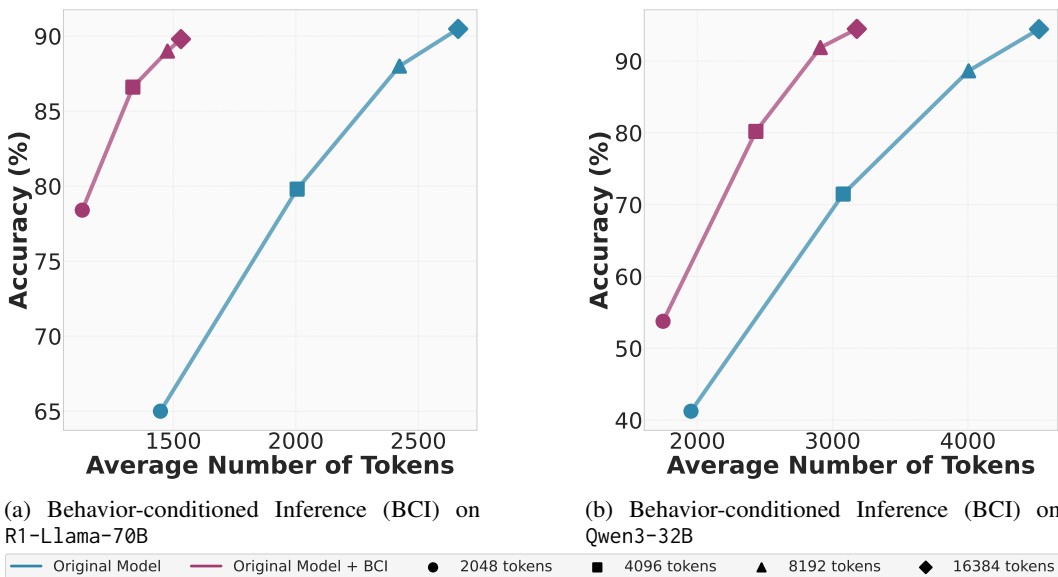

(a) Behavior-conditioned Inference (BCI) on `R1-Llama-70B`

(b) Behavior-conditioned Inference (BCI) on `Qwen3-32B`

Figure 2: **Behavior-conditioned Inference (BCI) for MATH**. Using behaviors from `R1-Llama-70B`, BCI is applied on two models - `R1-Llama-70B` and `Qwen3-32B`- while evaluating on the MATH-500 dataset (Hendrycks et al., 2021). The x-axis shows the average number of tokens produced per solution, the y-axis depicts the accuracy, and each point on the line corresponds to a particular hard token budget which is enforced during inference as shown in the legend. BCI achieves superior token efficiency producing answers with similar or improved accuracy while utilizing far fewer tokens than the base models.

## 5 EXPERIMENTAL RESULTS

This section presents experimental results for each use-case described in section 4. Unless otherwise specified, the decoding temperature is set to 0.6 and top–p is set to 0.95.

### 5.1 BEHAVIOR-CONDITIONED INFERENCE FOR EFFICIENT MATHEMATICAL REASONING

For the first use-case, BCI is applied to the MATH (Hendrycks et al., 2021) and AIME–24/25 (MAA, 2024; 2025) datasets. Two open-source reasoning LLMs - DeepSeek-R1-Distill-Llama-70B (`R1-Llama-70B`) (Guo et al., 2025) and Qwen3-32B (`Qwen3-32B`) (Bai et al., 2023) - are used as `Student` candidates.

`R1-Llama-70B` is employed as the `Metacognitive Strategist`. More dataset-specific details are provided in the following paragraphs.

**MATH Dataset** The behavior handbook is curated using a random sample of 1k questions from the MATH training set. The reasoning traces, reflections, and the behaviors are generated with a token budget of 8,192. All reasoning traces are used for curating the behavior handbook regardless of correctness. Here is the obtained distribution of behaviors per MATH subject - Algebra: 113, Prealgebra: 144, Counting: 128, Intermediate Algebra: 107, Geometry: 95, Number Theory: 107, and Precalculus: 91. Appendix Table A.2 presents a few example behaviors for each subject.

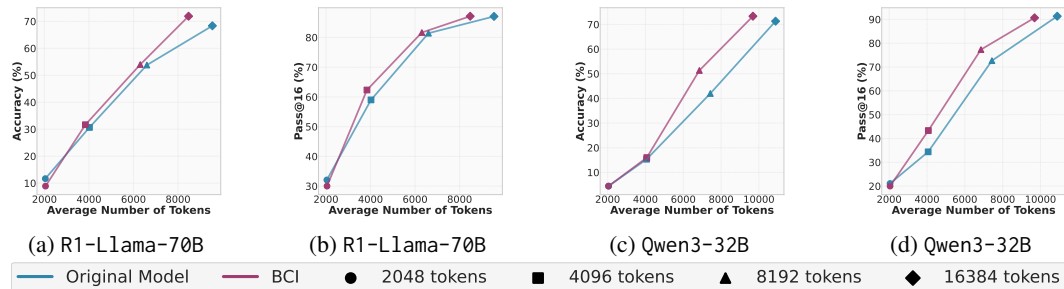

(a) R1-Llama-70B    (b) R1-Llama-70B    (c) Qwen3-32B    (d) Qwen3-32B

Figure 3: **Behavior-conditioned Inference (BCI) for AIME–24**. This figure presents results for the AIME–24 dataset. The accuracy is averaged across 80 runs. Pass@16 is averaged across 5 different sets of 16 runs each. The x-axis denotes the average number of tokens generated across all solutions. Each point on the line indicates a given token-budget which is enforced during generation as shown in the legend. The proposed approach improves the token efficiency of the generated solutions achieving superior or competitive performance while producing significantly lesser number of tokens.

**Baseline**    The baseline in this case runs normal inference on the same LLM by simply prompting it with the given problem and asking it to reason and produce the final answer in boxed format.

During inference, relevant behaviors are retrieved from the behavior handbook for a given question based on topic-matching. The evaluation results on the MATH-500 set are presented in Figure 2. The mean accuracy across 5 seeds is reported in the plot. The proposed BCI approach, achieves similar or improved accuracy while utilizing fewer tokens than the original model. Secondly, the performance still scales with increasing token-budget thus the proposed approach does not affect the model's existing capabilities in unwanted ways.

A few example reasoning traces output by the R1-Llama-70B during BCI are presented in Table A.1. Only those parts of the trace which utilize behaviors are shown. Some behaviors encode core mathematical concepts such as behavior_total_outcomes and behavior_inclusion_exclusion while others encode more general problem solving strategies such as behavior_estimate_reasonableness and behavior_translate_verbal_to_equation.

**AIME Datasets**    BCI is evaluated on AIME-24 (MAA, 2024) and AIME-25 (MAA, 2025). The behavior handbook for these benchmarks is derived from past AIME-22/23 exams (30 questions each, 60 total). Using R1-Llama-70B as the Metacognitive Strategist, 16 reasoning traces are generated per question at a 16,384-token budget. Reflections and behaviors are also generated at this budget, producing a final set of 1,457 behaviors (see Table A.3 for examples).

During inference, relevant behaviors are retrieved via embedding search. Both the question text and the 1,457 behaviors (with instructions) are encoded using the BGE-M3 sentence transformer (Chen et al., 2024). A FAISS index (Douze et al., 2024) built over these embeddings returns the top-$k$ candidates, with $k = 40$ in our experiments.

Results are shown in Figures 3 and 4. Accuracy is averaged across 80 runs per question, and pass@16 is computed over 5 sets of 16 responses each. BCI delivers more token-efficient solutions, achieving higher or comparable accuracy while using fewer tokens.

**Efficiency Considerations**    The proposed BCI approach enables models to produce fewer output tokens without sacrificing performance and scalability. This reduction in generation length has the potential to substantially lower the cost of inference compared to conventional prompting strategies. While the proposed method involves a larger number of input tokens due to the inclusion of retrieved behaviors, this overhead is mitigated by two key factors. First, the input representations of behaviors can be pre-computed and reused across different queries, amortizing the cost over multiple inferences. Second, there is no autoregressive generation required on the input side, which makes processing these tokens much faster. Moreover, in many proprietary model APIs, input tokens are billed at a lower rate than output tokens, making the proposed approach even more attractive from a cost-efficiency standpoint.

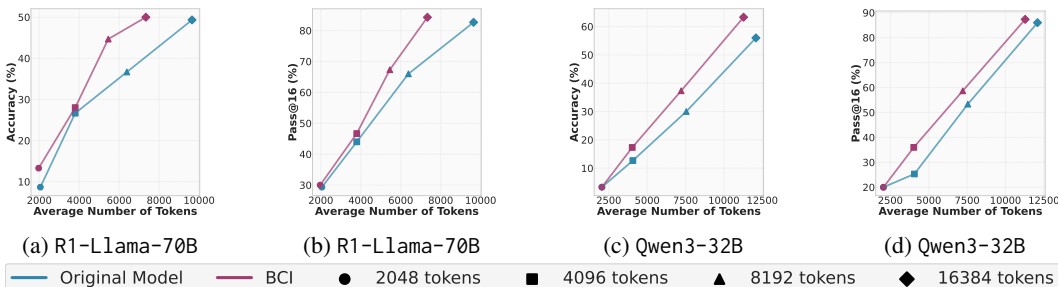

(a) R1-Llama-70B     (b) R1-Llama-70B     (c) Qwen3-32B     (d) Qwen3-32B

Figure 4: **Behavior-conditioned Inference (BCI) for AIME–25**. This figure presents results for the AIME–25 dataset using behavior list extracted using AIME–22 and 23. The accuracy is averaged across 80 runs. Pass@16 is averaged across 5 different sets of 16 runs each. The x-axis denotes the average number of tokens generated across all solutions. Each point on the line indicates a given token-budget which is enforced during generation. The proposed approach improves the token efficiency of the generated solutions achieving superior or competitive performance while producing significantly lesser number of tokens.

## 5.2 SELF-IMPROVEMENT USING BEHAVIORS

This section evaluates a model's self-improvement capabilities by using it as the `Metacognitive Strategist` and the `Student` at the same time. `R1-Llama-70B` is used for both the roles.

**Critique-and-Revise Baseline** The baseline in this experiment uses an LLM for self-improvement by directly prompting it to critique-and-revise its own past reasoning trace. Concretely, given a question $Q$, the model first produces an initial reasoning trace $R_1$ ($Q \rightarrow R_1$). This reasoning trace along with the original question is then fed back into the model prompting it to generate an improved trace $R_2$ that corrects errors or extends incomplete reasoning ($(Q, R_1) \rightarrow R_2$). This baseline is called the *critique-and-revise* baseline. In

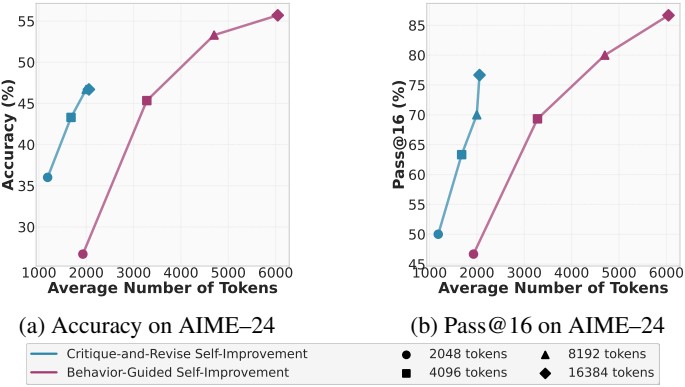

(a) Accuracy on AIME–24     (b) Pass@16 on AIME–24

Figure 5: **Self Improvement** Self-Improvement. Comparison of critique-and-revise ($(Q, R_1) \rightarrow R_2$) vs. behavior-guided ($(B, Q) \rightarrow R_2$) on AIME-24. Results are averaged over 80 runs (Pass@16 over 5×16 runs). Behavior-guided produces more tokens and is less token-efficient but achieves markedly higher accuracy and Pass@16, especially at larger token budgets. All runs use `R1-Llama-70B` with a 2,048-token initial trace and revision budgets from 2,048 to 16,384

this experiment, the token budget for $R_1$ is set to 2,048 and that of $R_2$ is varied from 2,048 to 16,384.

**Behavior-guided variant** In the behavior-guided variant, behaviors are curated from the original reasoning traces $R_1$ which are generated at a token budget of 2,048. Following this, each step in the behavior curation pipeline is executed at a token budget of 2,048. A total of 16 reasoning traces are generated per question which are used by the `Metacognitive Strategist` to curate a behavior handbook for that question. Finally, these behaviors are fed back into the model to generate improved reasoning traces at budgets ranging from 2,048 to 16,384 ($(B, Q) \rightarrow R_2$).

**Results** Figure 5 shows three key patterns. (1) Accuracy: Behavior-conditioned revisions ($(B, Q) \rightarrow R_2$) outperform critique-and-revise ($(Q, R_1) \rightarrow R_2$) at nearly all budgets, with the gap widening as more tokens are allowed. (2) Scaling: Behavior-guided self-improvement maintains the smooth test-time scaling of `R1-Llama-70B`, while critique-and-revise struggles to leverage larger

budgets. (3) Efficiency: Unlike earlier settings, the behavior-guided approach is less token-efficient here, producing longer outputs than the baseline.

## 5.3 SUPERVISED FINE-TUNING WITH BEHAVIORS

Behavior-conditioned supervised fine-tuning (BC-SFT) tries to incorporate good behaviors into the model's parameters. In this setting, R1-Llama-70B is the `Metacognitive Strategist` which generates behaviors as well as the `Teacher` which generates behavior-conditioned responses for training. The following 4 candidates are used as `Student` models which are fine-tuned: `Qwen2.5-14B` (Qwen et al., 2025), `Qwen2.5-32B-Instruct` (Qwen et al., 2025), `Qwen3-14B` (Bai et al., 2023), and `Llama-3.1-8B` (Dubey et al., 2024).

**Dataset construction.** The problems from the S1 dataset (Muennighoff et al., 2025) are used to create the training datasets used in this experiment. For each problem $Q_i$, the `Metacognitive Strategist` (R1-Llama-70B) is used to create a set of behaviors using the pipeline from Section 3. A single reasoning trace $R_i$ is used per problem to create the behavior handbook. Each step in the pipeline is run at a token budget of 14,000. Next, using BCI, a dataset of behavior-conditioned responses is curated using the `Teacher` (R1-Llama-70B) model for training the `Student` models.

$$\mathcal{D}_{\text{BC}} = \{(Q_0, \hat{R}_0), \ldots, (Q_n, \hat{R}_n)\}.$$

where, $\hat{R}_i$ is the behavior-conditioned response generated from the `Teacher`. The behaviors are retrieved from the behavior handbook based on question matching i.e. behaviors generated from a given question are used in-context for generating a response to that question. For the baseline, the `Student` models are trained on the corpus of the original reasoning traces generated from the `Teacher` with normal inference.

$$\mathcal{D}_{\text{SFT}} = \{(Q_0, R_0), \ldots, (Q_n, R_n)\}.$$

Importantly, behaviors are *not* provided as input during fine-tuning on $\mathcal{D}_{\text{BC}}$, or in-context during test-time; any benefit at test time therefore reflects knowledge distilled into the parameters themselves.

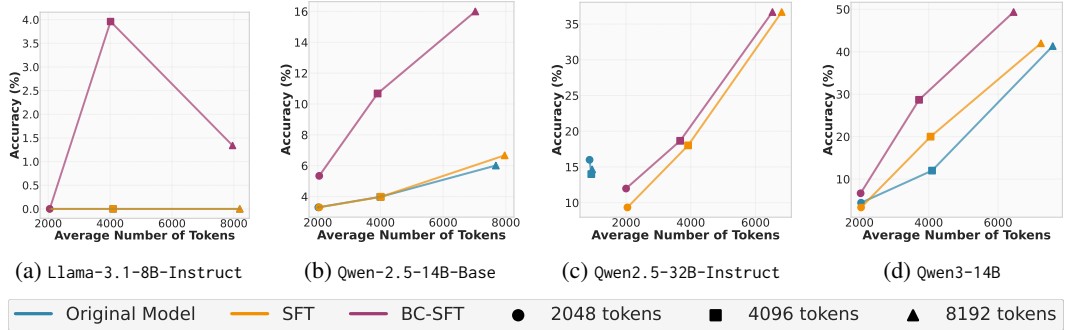

(a) `Llama-3.1-8B-Instruct`  (b) `Qwen-2.5-14B-Base`  (c) `Qwen2.5-32B-Instruct`  (d) `Qwen3-14B`

Figure 6: **SFT Experiment: AIME–24**. Each panel plots accuracy (%) versus the average number of generated tokens for one base model. Three variants are evaluated: *Original* (no additional training), *SFT* (fine-tuned on the original corpus $\mathcal{D}_{\text{SFT}}$), and *BC-SFT* (fine-tuned on behavior-conditioned SFT corpus $\mathcal{D}_{\text{BC-SFT}}$). The accuracy is averaged across 80 runs. Each point on the line indicates a given token-budget which is enforced during generation as shown in the legend. The BC-SFT fine-tuned model consistently achieves superior performance compared to the other two variants across all token budgets while also being more token efficient.

**Training setup and evaluation.** The same setting as that of Muennighoff et al. (2025) is adapted for this experiment. Each model is fine-tuned with a 16,384-token context window. At inference only the question is supplied to the model and decoding is performed with budgets from 2,048 to 8,192 tokens.

**Results.** Figures 6 and 7 show AIME-24/25 accuracy versus token budget (2,048–8,192) for all four base models (*Original*, *SFT*, and *BC-SFT*). The key result is that BC-SFT models are both more token-efficient and consistently more accurate than baselines. Unlike standard SFT, BC-SFT is especially effective at turning non-reasoning models (e.g., `Qwen2.5-14B-Base`, `Qwen2.5-32B-Instruct`) into strong reasoning models. This contrasts with prior in-context BCI results, where improvements came mainly from efficiency at large budgets ($> 8,192$ tokens). Here, BC-SFT yields genuine quality gains: models fine-tuned on behavior-conditioned traces outperform those trained on original traces even under the same budget, showing that the supervision imparts useful reasoning behaviors rather than just brevity. To probe whether these gains could be attributed simply to better answer correctness in the training data, the responses in $\mathcal{D}_{\text{SFT}}$ and $\mathcal{D}_{\text{BC}}$ were evaluated against the S1 reference answers, obtaining 42.7% and 44.4% accuracy, respectively—a negligible gap that cannot explain the downstream performance deltas. Nevertheless, models trained on $\mathcal{D}_{\text{BC}}$ achieve markedly superior AIME performance (see especially panel (b) in Figures 6 and 7, `Qwen2.5-14B-Base`, where accuracy scales sharply with budget under BC-SFT while the plain SFT model improves only modestly). Together, the results suggest that behavior-conditioned supervision injects intermediate reasoning skills into the model, enabling more accurate and efficient problem solving than conventional SFT.

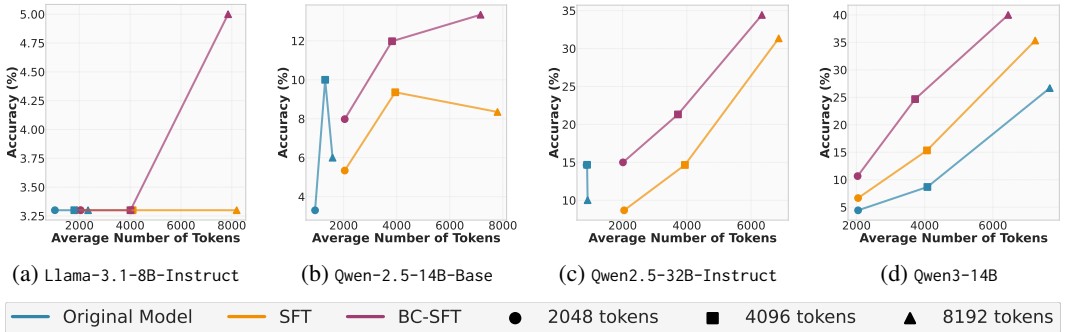

(a) `Llama-3.1-8B-Instruct`  (b) `Qwen-2.5-14B-Base`  (c) `Qwen2.5-32B-Instruct`  (d) `Qwen3-14B`

Figure 7: **SFT Experiment: AIME–25**. Each panel plots accuracy (%) versus the average number of generated tokens for one base model. Three variants are evaluated: *Original* (no additional training), *SFT* (fine-tuned on the original corpus $\mathcal{D}_{\text{SFT}}$), and *BC-SFT* (fine-tuned on behavior-conditioned SFT corpus $\mathcal{D}_{\text{BC-SFT}}$). The accuracy is averaged across 80 runs. Each point on the line indicates a given token-budget which is enforced during generation as shown in the legend. The BC-SFT fine-tuned model consistently achieves superior performance compared to the other two variants across all token budgets while also being more token efficient.

## 6 CONCLUSION AND LIMITATIONS

This work introduces a mechanism for large language models to distill recurring reasoning patterns into concise *behaviors*. By storing and retrieving these behaviors, models avoid re-deriving the same intermediate steps, instead recalling relevant behaviors and spending tokens on new reasoning. Across three settings—behavior-conditioned inference, behavior-guided self-improvement, and behavior-conditioned supervised fine-tuning—the approach yields consistent gains in both accuracy and token efficiency on challenging math benchmarks.

Beyond mathematics, the framework is model- and domain-agnostic, with potential applications in programming, theorem proving, scientific reasoning, and open-ended dialogue. Limitations remain: in BCI, retrieved behaviors are fixed once provided, whereas ideally the model would retrieve new behaviors on the fly, perhaps by learning to query the behavior handbook as a "tool." More broadly, scaling the framework will require (1) curating large cross-domain behavior libraries, and (2) performing large-scale SFT with behavior-rewritten corpora, both to improve smaller models and to bootstrap the behavior-generation process itself.

Overall, turning slow chains of thought into fast, reusable behaviors enables more efficient and scalable reasoning, pointing toward LLMs that not only solve problems but also remember how.

## 7 REPRODUCIBILITY STATEMENT

We have taken several steps to ensure the reproducibility of our results. The full behavior curation pipeline, including prompts for solution, reflection, and behavior extraction, is described in Section 3 and detailed in Appendix Figures A.1 and A.2. Experimental setups for behavior-conditioned inference, behavior-guided self-improvement, and behavior-conditioned supervised fine-tuning are explained in Section 4 and evaluated in Section 5, with dataset-specific details provided for MATH and AIME benchmarks. Hyperparameters such as decoding settings and token budgets are explicitly noted in Section 5. Example behaviors curated from the datasets are included in Appendix Tables A.2, A.3, A.4, and A.5. All datasets used (MATH (Hendrycks et al., 2021) and AIME (MAA, 2024; 2025)), are publicly available.

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

# APPENDIX

| Solution Prompt | Reflection Prompt |
|---|---|
| Please reason step by step and put the final answer in [____]

Problem: \<problem\> | Here is the definition of a behavior:
• A behavior is a note or skill to keep in mind while solving math problems.
• It can be a strategy, a trick, or a technique.
• It can also be a general rule or a common sense principle.
• A behavior is not a solution to the problem, but it can be used to solve the problem.

For example - if the problem is "Find the area of a circle with radius 4", one useful behaviour could be {"behavior_area_of_circle": area of a cirle is pi*r^2}.

Given a problem and the corresponding solution, reflect and critique the solutions along the following dimensions: |

| Behavior Prompt |
|---|
| \<reflection prompt\>

\<reflection\>

Now, given this reflection generate a list of behaviors and corresponding instructions/ explanations in json format. Each behavior should be a single line, and the format is "behavior_[name]: [description]". The list should be in json format, and each behavior should be a key-value pair, where the key is the behavior name and the value is the description. |

1. Correctness Analysis: Is the answer mathematically correct? Are there calculation errors? Is the reasoning logically sound? Are all steps properly justified? What specific mistakes were made?
2. Missing Behaviors Analysis: What behaviors should have been used but weren't? Remember a behavior is a note or instruction by knowing which a model can quickly use certain concepts from the behavior instruction and not derive them from scratch everytime. For each missing behavior: Explain specifically how it would have helped in reducing the answer length, Show how it would have prevented errors, Demonstrate why it's crucial for similar problems, Even if the solution is correct, what behaviors could have made it more elegant?
3. New Behavior Suggestions: Suggest specific new behaviors that will help with similar problems. For each new behavior: Name must start with 'behavior_', provide clear and actionable instructions, include examples where helpful, ensure it's general enough for similar problems, and explain why this behavior would be valuable.

Figure A.1: Prompts used for extracting behaviors from thoughts are listed in this Figure. **Solution Prompt** is used to map the questions to solutions containing reasoning traces. Next, the **Reflection Prompt** is employed to generate a reflection for the solution followed by using the **Behavior Prompt** to generate the behaviors.

| Two fair, 6-sided dice are thrown. What is the probability that the product of the two numbers is a multiple of 5? Express your answer as a common fraction. | Remmy wants to divide 10 by $\frac{2}{3}$, but he cannot remember how to do that. By what number should he multiply 10 to get the answer? | Hillary has eleven coins, all dimes and nickels. In total, the coins are worth 75 cents. How many nickels does she have? |
|---|---|---|
| Alright, so I need to find the probability that the product of two fair 6-sided dice is a multiple of 5. Hmm, okay.
...
...
That would be 6 (for the first die) multiplied by 6 (for the second die), which gives us 36 total possible outcomes. I think this is an application of the behavior_total_outcomes behavior.
...
...
There are two scenarios where the product is a multiple of 5:

  1. The first die is a 5, regardless of what the second die shows.

  2. The second die is a 5, regardless of what the first die shows.

Wait a second — if I just add these two scenarios together, I might be double-counting the case where both dice are 5. So, I need to adjust for that overlap to avoid counting it twice.
This sounds like the behavior_inclusion_exclusion principle, where I add the probabilities of each event and then subtract the probability of their intersection.... | Alright, let me try to figure out how to solve this problem. So, Remmy wants to divide 10 by 2/3, but he's forgotten how to do that
...
...
As per the behavior behavior_reciprocal, I know that dividing by a fraction is the same as multiplying by its reciprocal.
...
...
So, my initial answer of 15 seems correct. Another way to think about it is using the behavior behavior_estimate_reasonableness. If I estimate, dividing 10 by a number less than 1 should give me a number larger than 10. Since 2/3 is about 0.666, dividing 10 by 0.666 should be around 15, which matches my previous result.... | Alright, so Hillary has eleven coins, and they're all dimes and nickels. The total value is 75 cents. I need to figure out how many nickels she has. Let me break this down step by step.
First, I'll use the behavior behavior_translate_verbal_to_equation to convert the problem into mathematical equations. Let's let $d$ represent the number of dimes and $n$ represent the number of nickels.
I know two things: the total number of coins and the total value. So, I can write two equations. The first equation is about the number of coins:

$$d + n = 11$$

This is because she has eleven coins in total, and they're all dimes and nickels.
The second equation is about the total value. Since each dime is worth 10 cents and each nickel is worth 5 cents, the total value equation is:
$$10d + 5n = 75$$
.... |

Table A.1: **Behavior Conditioned Inference Examples** This table illustrates how R1-Llama-70B uses behaviors to ease problem solving. The behaviors used are highlighted in blue.

```
┌─────────────────────────────────────────────────────────────────────┐
│                 Behavior-Conditioned Inference Prompt                 │
├─────────────────────────────────────────────────────────────────────┤
```
A behavior is a note or skill to keep in mind while solving math problems. It can be a strategy, a trick, or a technique. It can also be a general rule or a common sense principle. The behavior is not a solution to the problem, but it can be used to solve the problem.

Here is a list of behaviors:

{behavior_1} : <instruction_1>
{behavior_2}: <instruction_2>
{behavior_3}: <instruction_3>
                ....

Now, solve the following math problem efficiently and clearly. You can use any of the behaviors above to solve the problem.

In your reasoning, when you use a behavior explicitly refer to the behaviors when you use them.

Please reason step by step and put the final answer in \\boxed{{}}

Figure A.2: Prompt used for behavior-conditioned inference (BCI).

| Subject | Behavior Name | Behavior Instruction |
|---------|---------------|----------------------|
| Algebra | behavior_use_systematic_verification | When performing multi-step calculations, use a systematic approach such as creating a table to track each step's results, ensuring each step is verified before proceeding to the next. |
| | behavior_simplify_after_substitution | After substituting variables, simplify the resulting expression as much as possible to enhance clarity and correctness. |
| | behavior_recognize_algebraic_patterns | Look for common patterns such as perfect squares, cubes, or factorable forms to simplify problem-solving and blueuce errors. |
| | behavior_always_rationalize_denominators | Always rationalize denominators to present answers in simplest form, avoiding radicals in the denominator. |
| | behavior_check_units | Verify all units are consistent before performing calculations; e.g., ensure time is in years if the rate is annual. |
| Prealgebra | behavior_assign_variables_wisely | Use descriptive variable names that match their meaning in the problem to avoid confusion. |
| | behavior_prioritize_calculation_order | Follow the correct order of operations to maintain accuracy. |
| | behavior_translate_words_to_equation | Convert phrases into math expressions; e.g., "one and one-half of a number" becomes $(3/2)x$. |
| | behavior_understand_independent_events | Identify independent events and apply the multiplication rule accordingly. |
| | behavior_check_for_zero_exponents | Confirm non-zero base before applying the rule that anything to the zero power is 1. |
| Counting and Probability | behavior_counting_principle | Multiply the number of choices for each independent step to get total possibilities. |
| | behavior_inclusion_exclusion_principle | Use $P(A \cup B) = P(A) + P(B) - P(A \cap B)$ to avoid double-counting. |
| | behavior_complementary_probability | Use the complement when it's simpler than calculating direct probability. |
| | behavior_consider_symmetry | Account for symmetries in arrangements (e.g., rotations) to blueuce unique cases. |
| | behavior_binomial_distribution | Use binomial formula $P(k) = C(n, k)p^k(1-p)^{n-k}$ for fixed, independent binary trials. |
| Geometry | behavior_use_shoelace_formula | Use the shoelace formula to find polygon area from coordinates. |
| | behavior_use_trigonometric_relations | Apply Law of Sines, Cosines, or dot products for angle-side relationships. |
| | behavior_analyze_quadrant_signs | Use sign rules for trig functions based on angle quadrant. |
| | behavior_use_cross_product_for_area | Use cross product to compute area of polygons using vectors. |
| | behavior_check_perpendicularity_with_dot_product | Dot product of zero implies vectors are perpendicular. |
| Precalculus | behavior_apply_power_blueuection_formulas | Use power-blueuection identities to simplify high powers of sine/cosine. |
| | behavior_use_tangent_addition_formula | Use $\tan(A + B)$ formula to simplify expressions involving tangent sums. |
| | behavior_convert_to_polar_form | Use polar form for complex operations like exponentiation. |
| | behavior_magnitude_calculation | Square each component, sum them, and take square root for vector magnitude. |
| | behavior_check_orthogonality | Dot product of 0 means vectors are orthogonal. |
| Number Theory | behavior_check_prime_factors | Confirm prime factorization is complete and correct for each number. |
| | behavior_convert_between_bases | Expand each digit's place value step-by-step when converting bases. |
| | behavior_factor_common_terms | Factor out shablue terms early to simplify later computation. |
| | behavior_recognize_repeating_patterns | Use repeating digit patterns to aid in factoring or GCD steps. |
| | behavior_systematically_record_remainders | Track remainders carefully in Euclidean algorithm or base conversion. |
| Intermediate Algebra | behavior_apply_vieta_formulas | Use Vieta's formulas to relate polynomial roots to coefficients. |
| | behavior_recognize_telescoping_patterns | Identify cancellation patterns in fraction products to simplify. |
| | behavior_use_calculus | For optimization, derive the function, find critical points using calculus. |
| | behavior_recognize_conjugate_pairs | Know complex roots appear in conjugate pairs in real-coefficient polynomials. |
| | behavior_triangle_inequality_check | Ensure sum of any two sides exceeds the third to form a triangle. |

Table A.2: Examples of behaviors curated from the MATH Dataset (Hendrycks et al., 2021)

| Behavior Name | Behavior Instruction |
|---|---|
| behavior_permutation_with_restrictions | Use permutation techniques with restrictions to count valid configurations in grid problems. |
| behavior_rhombus_incircle_distances | In a rhombus, the sum of distances from any point on the incircle to two opposite sides is equal to twice the inradius. |
| behavior_opposite_sides_sum | In a parallelogram, the sum of distances from any interior point to two opposite sides is constant and equal to the height between those sides. |
| behavior_periodicity | Recognize repeating patterns in remainders to efficiently count valid numbers within a range. |
| behavior_remainder_distinctness | Always check that remainders from different moduli are distinct, especially when moduli are not pairwise coprime. |
| behavior_dynamic_programming | Use dynamic programming to break down complex problems into smaller subproblems and solve them recursively. |
| behavior_ptolemy_theorem | Apply Ptolemy's theorem to cyclic quadrilaterals to find relationships between sides and diagonals, such as $AC \cdot BD = AB \cdot CD + AD \cdot BC$. |
| behavior_concurrent_intersections | For each point where $k$ lines intersect, adjust the intersection count by subtracting $\left( \binom{k}{2} - 1 \right)$ to account for overlapping. |
| behavior_volume_ratio_parallelepiped | Calculate the volume ratio of two noncongruent parallelepipeds by considering the angles between their edges and applying the volume formula for rhombohedrons. |
| behavior_vector_analysis | Use vector methods or barycentric coordinates to simplify geometric problems involving angles and concurrency. |
| behavior_fibonacci_non_consecutive | Use the Fibonacci sequence to count subsets with no two consecutive elements; for a set of size $n$, the count is the $(n+2)$th Fibonacci number. |
| behavior_legendre_formula | Use Legendre's formula to find the exponent of a prime in the factorization of a factorial. |
| behavior_modular_overlap | Use systematic methods like the Chinese Remainder Theorem and inclusion-exclusion to detect and count overlaps in modular conditions. |
| behavior_distance_from_point_to_line | Calculate the distance from a point to a line using $\frac{|Ax+By+C|}{\sqrt{A^2+B^2}}$ to verify tangency. |
| behavior_perpendicular_lines | Recognize that radii to tangent points are perpendicular to the tangent line, aiding in finding slopes and equations. |
| behavior_pythagorean_theorem | Use the Pythagorean theorem $a^2 + b^2 = c^2$ to find the length of a side in a right triangle. |
| behavior_diophantine_simplification | Simplify Diophantine equations by factoring out common terms or recognizing patterns in coefficients. |
| behavior_tangent_properties | Recall that the radius of a circle is perpendicular to the tangent at the point of tangency, aiding in determining the center. |
| behavior_mixtilinear_incircle_radius | Calculate the radius of a mixtilinear incircle using: $r_A = \frac{R \cdot \sin^2\left(\frac{A}{2}\right)}{1 - \sin\left(\frac{A}{2}\right)}$ for ex-mixtilinear, and $r_A = \frac{R \cdot \sin^2\left(\frac{A}{2}\right)}{1 + \sin\left(\frac{A}{2}\right)}$ for internal mixtilinear circles. |
| behavior_proportion_analysis | When dealing with proportions before and after an event, set up equations based on initial and final proportions and solve methodically. |

Table A.3: Examples of behaviors curated from AIME 22-23 set.

| Behavior Name | Behavior Instruction |
|---|---|
| behavior_grid_assignment_counting | Calculate the number of configurations for grid problems by considering $2^{(\text{rows + columns})}$, adjusted for constraints. |
| behavior_use_properties_of_exponents | Use properties of exponents to combine terms efficiently when simplifying expressions. |
| behavior_count_rectangles_in_regular_polygons | When counting rectangles in a regular polygon, consider perpendicular diagonal pairs and use symmetry to avoid overcounting. For a regular $n$-gon, identify all sets of four vertices forming rectangles by checking perpendicularity and equal lengths. |
| behavior_domain_restriction | When dealing with functions like $\tanh$, which have restricted ranges, always consider their effect on derived variables to set valid domains for optimization. |
| behavior_consistent_signage | Maintain consistent sign conventions, especially when solving geometry problems in coordinate space. |
| behavior_convert_log_to_exponential | Convert logarithmic equations to exponential form to simplify variable relationships. |
| behavior_transcendental_equation_handling | Recognize when equations may not admit algebraic solutions and consider numerical or graphical methods instead. |
| behavior_fermats_little_theorem | Apply Fermat's Little Theorem: if $p$ is prime and $n$ is not divisible by $p$, then $n^{p-1} \equiv 1 \mod p$. |
| behavior_tangent_segment_length | Calculate tangent length from a point to a circle using $\sqrt{d^2 - r^2}$, where $d$ is the distance to the center and $r$ is the radius. |
| behavior_time_conversion | Always convert hours to minutes or vice versa to maintain consistent units throughout time calculations. |
| behavior_inradius_calculation | Use $r = \text{Area}/s$ to find the inradius of a triangle, where $s$ is the semi-perimeter. |
| behavior_calculus_optimization | Use calculus to find extrema: compute the derivative, set it to zero, solve for critical points, and verify using the second derivative or sign changes. |
| behavior_astroid_properties | Recognize the astroid formed by envelopes from $(p, 0)$ to $(0, q)$ satisfying $p^2 + q^2 = 1$. The astroid is given by $x^{2/3} + y^{2/3} = 1$, useful for tangency problems. |
| behavior_counting_symmetric_pairs | In regular polygons, count figures by analyzing all symmetric vertex pairs and step sizes. For a dodecagon, consider both $k$ and $n - k$ step sizes. |
| behavior_systematic_enumeration | Enumerate all cases methodically while honoring constraints to avoid missed or duplicate solutions. |
| behavior_apply_intercept_theorem | In problems with parallel lines cutting segments, use the intercept theorem to relate segment lengths proportionally. |
| behavior_law_of_cosines | Use $c^2 = a^2 + b^2 - 2ab\cos(C)$ to connect side lengths and angles in triangles. |
| behavior_mode_implications | Use the fact that the mode appears more frequently than other values to structure the dataset accordingly. |
| behavior_hyperbolic_identities | Use identities like $\cosh^2 \theta - \sinh^2 \theta = 1$ when parametrizing or manipulating hyperbolas. |
| behavior_use_cyclic_properties | In cyclic quadrilaterals, apply Ptolemy's Theorem or Power of a Point to relate sides and diagonals. |

Table A.4: Example of behaviors obtained from AIME-24 for the self-improvement experiment

| Behavior Name | Behavior Instruction |
|---|---|
| behavior_distance_from_point_to_line | Calculate the distance from a point to a line using $\frac{|ax+by+c|}{\sqrt{a^2+b^2}}$. |
| behavior_arrhenius_formula | Apply the Arrhenius formula $k = A \cdot \exp\left(-\frac{E}{k_B T}\right)$ for temperature-dependent reaction rates. |
| behavior_gradient_check | Verify that the gradient (sum of unit vectors) is zero at a potential minimizer to confirm it's the geometric median. |
| behavior_iterative_methods | Use iterative algorithms like the Weiszfeld method to approximate the geometric median when a closed-form solution is not feasible. |
| behavior_dipole_potential_inside | For dipole-like surface charge distributions, the electric potential inside a spherical shell behaves like $r \cos \theta$. |
| behavior_permutations_with_duplicates | Compute distinct permutations with repeated elements using $\frac{n!}{n_1! n_2! \cdots n_k!}$. |
| behavior_area_parallelogram_complex | Compute the area of a parallelogram formed by complex numbers using the imaginary part of $z_1 \cdot \overline{z_2}$. |
| behavior_divergent_series_handling | Be cautious with divergent series; sometimes combinations of divergent sums can converge via cancellation. |
| behavior_transposition_cycles | The minimum number of transpositions to sort a permutation is $n - c$, where $c$ is the number of cycles. |
| behavior_kl_divergence_relations | The chi-square statistic is twice the leading term in the Taylor expansion of the Kullback–Leibler divergence. |
| behavior_gcd_usage | Use the GCD to break down integers and solve linear Diophantine equations like $ax + by = d$. |
| behavior_mirror_formula | Use the mirror formula $\frac{1}{f} = \frac{1}{v} + \frac{1}{u}$ for image distances in spherical mirrors. |
| behavior_kinetic_energy_eV | Mean kinetic energy of a gas molecule is $E = \frac{3}{2}kT$, where $k = 8.617 \times 10^{-5}$ eV/K. |
| behavior_midpoint_calculation | Find the midpoint of a segment using the average of the endpoints' coordinates. |
| behavior_prime_exponent_independence | Treat each prime factor independently when handling divisibility conditions; calculate probabilities separately then combine. |
| behavior_cauchy_variation | Add a constant to $f(x + y) = f(x) + f(y) + C$ and define $g(x) = f(x) + C$ to convert to a Cauchy equation. |
| behavior_lambert_w_recognition | Recognize that equations like $xe^x = k$ can be solved using the Lambert W function: $x = W(k)$. |
| behavior_permutation_cycles | Consider cycle structure to simplify permutation counting, especially with requiblue or forbidden cycle lengths. |
| behavior_cubing_and_cube_roots | Recognize cubes or cube roots to simplify expressions, especially in volume-related problems. |
| behavior_lagrange_multipliers | Use Lagrange multipliers for constrained optimization by forming the Lagrangian and solving the system from partial derivatives. |
| behavior_lorentz_factor_approximation | Use $\sqrt{1 - v^2/c^2} \approx 1 - \frac{v^2}{2c^2}$ to approximate time dilation effects at small velocities. |

Table A.5: Examples of behaviors curated on the S1 dataset (Muennighoff et al., 2025) for the SFT experiment.

