# OpenReview forum: "Metacognitive Reuse: Turning Recurring LLM Reasoning Into Concise Behaviors"
_ICLR.cc/2026/Conference — Submitted to ICLR 2026_

### Official Review · Reviewer_ohC1 · 2025-10-18

**Soundness:** 2
**Presentation:** 3
**Contribution:** 2
**Rating:** 2
**Confidence:** 4

**Summary:**

This work introduces a mechanism for LLMs to store and reuse recurring reasoning patterns into a "behavior handbook", so that they can avoid re-inventing / re-deriving the same intermediate thinking steps for new yet similar problem solving process. The approach shows promising gains in both accuracy and token efficiency on challenging math benchmarks.

**Strengths:**

S1. The idea is original and novel, which could bring some new insights into the area of reasoning paradigm designs.

**Weaknesses:**

W1. Limited improvement in **accuracy**.

The results reported in Sections 5.1 and 5.2 do not demonstrate a meaningful improvement in accuracy. For the behavior-conditioned inference (BCI) setting, Figure 3 indicates that BCI improves token efficiency compared to the naive baseline (i.e., solving the problem without behavioral guidance). However, it does not enhance the upper-bound performance of student models—particularly when the student model is weaker than the teacher (Figure 3 c, d). In the self-improvement study (Section 5.2), the critique-and-revise baseline performs even worse than the naive inference baseline introduced in Section 5.1, suggesting that the claimed improvement is achieved relative to an artificially weak baseline. As a result, the accuracy gains emphasized in the abstract and Section 5.2 appear less meaningful and less convincing.

W2. Debatable token efficiency analysis.

The evaluation of token efficiency is not entirely convincing. Although Figures 2 and 3 report improved output token efficiency, the proposed method introduces a large number of additional input tokens (e.g., Lines 309–310 mention inserting 40 behaviors into the input prompt, which could contribute hundreds or even thousands of extra tokens). This significantly offsets the apparent token savings. While these tokens are not generated autoregressively (as discussed in Line 321), they still impose computational overhead due to the quadratic cost with respect to sequence length. The paper should therefore analyze the actual computational cost associated with the extended prompts—such as total token counts, inference latency, or wall-clock computation time—to provide a more balanced efficiency assessment.

W3. Insufficient validation set size and generalization analysis.

The validation experiments rely primarily on AIME-24 and AIME-25, which each contain only 30–40 test samples. Such a small evaluation set lacks statistical significance and makes the results highly sensitive—for instance, solving just one additional problem can yield a 2.5–3.3 % increase in reported accuracy. Consequently, the results are insufficient to demonstrate the generalization ability of the proposed method. It remains unclear whether the observed gains generalize beyond this narrow domain or depend on specific characteristics of AIME-style problems. The authors are encouraged to include larger and more diverse benchmarks, such as GSM8K or MathVista, to validate the generality and robustness of their approach.

W4. Unsatisfactory SFT results.

The supervised fine-tuning (SFT) experiments are limited to smaller models (< 32 B parameters), and the observed improvements are modest compared to BCI performance on larger or equally sized models (e.g., Qwen3-32B in Figures 3 c and d). Although BC-SFT shows some gains over naive distillation baselines, it still exhibits substantial performance gaps relative to larger teacher models. Notably, for 32 B models, SFT underperforms BCI prompt engineering baseline (Figure 6 c vs. Figures 4 c, d), which undermines the claimed necessity and effectiveness of BC-SFT.

**Questions:**

Q1. Clarity and interpretation of behavior comparisons.

The comparison of behaviors is not very straightforward for readers. Introducing behavior guidelines could potentially introduce uncertain or inconsistent behaviors in the model’s responses. For instance, does the model sometimes wander by selecting diverse but unhelpful behaviors, thereby generating unnecessary tokens? How often does the original model “waste tokens” on repetitive recurring reasoning steps? It would be helpful if the authors could include qualitative case studies illustrating (1) how models without guidance tend to exhibit wasteful, recurring reasoning patterns, (2) how the proposed method helps guide the reasoning process more efficiently, and (3) whether the proposed method itself ever introduces new, unnecessary reasoning steps (e.g., being distracted by irrelevant behaviors).


Q2. Potential restriction of exploration space due to the behavior guideline.

Does the behavior guideline constrain the exploration space of the model? In the baseline setting, the model may have a broader but implicit strategy space (stored in their parameters) for problem solving. However, with the behavior book in place, does the model restrict its exploration to within the boundaries of the predefined behavior guidelines? It would be useful for the authors to provide examples showing whether the model can still apply a necessary problem-solving technique that is not covered in the behavior book.


Q3. Generalizability beyond math reasoning.

For mathematical reasoning, many of the behaviors or strategies in the behavior book may already exist in the model’s pretraining corpus—for instance, through exposure to established theorems or human-derived problem-solving heuristics. This could make deriving such behaviors relatively easy and potentially bias the results in favor of math tasks. How well does the proposed method generalize to domains that are less structured or verifiable, for example, creative writing or coding, where problem-solving heuristics are less well-defined?

---

### Official Review · Reviewer_gEgD · 2025-11-02

**Soundness:** 4
**Presentation:** 4
**Contribution:** 3
**Rating:** 8
**Confidence:** 4

**Summary:**

In this paper, the authors suggest that the intermediate CoT generated for multi-step problems can be reusable. Hence, they choose to extract the reusable 'behaviors' by allowing reasoning LLMs to perform metacognitive analysis of its prior reasoning traces. These behaviors are put together as the behavior handbook. Three settings of utilizing these behvaors are proposed, by behavior-conditioned inference, self-improvement and fine-tuning respectively. The empirical results suggest that the idea indeed can improve the reasoning efficiency as well as the effectiveness.

**Strengths:**

1. The paper is well written and easy to follow.
2. The idea of constructing a behavior handbook and reusing the former in-context is novel and reasonable.
3. The extensive experiments well demonstrate the superiority of the idea in multiple settings.

**Weaknesses:**

1. The authors are encouraged to perform the evaluation across different tasks. Right now, the experiments mainly focus on the mathematical reasoning.

**Questions:**

See the above 'Weaknesses'

---

### Official Review · Reviewer_u8Gy · 2025-11-02

**Soundness:** 2
**Presentation:** 3
**Contribution:** 2
**Rating:** 2
**Confidence:** 4

**Summary:**

This paper proposes to use LLMs to summarise solutions into "behaviours", or short instructions representing some solution strategies, and later reuse in different problems or with different LLMs. Given a set of problems, a "meta-cognitive strategist", which is essentially an LLM, is prompted to (1) generate solutions for these problems, (2) reflect on its own solutions, and (3) generate "behaviours" based on problem, solution, and reflections.

The paper demonstrates the usage of these discovered "behaviours" in three different settings, namely, (1) directly using behaviours as prompts (behaviour-conditioned inference), (2) as a source for LLM's self-improvement, and (3) used in SFT to improve reasoning. Empirical results on a few open-weight models demonstrate the efficacy of using behaviours.

**Strengths:**

1. Discovering reusable components in reasoning is an interesting idea. The paper has at least shown it works for some math problems.

2. The paper demonstrates different usages of "behaviours", which shows that the method could be useful for different application scenarios. However, in my view this is not empirically verified.

**Weaknesses:**

1. Baseline for BCI is weak. Line 288 of the paper says that the baseline is just LLM running normal inference, which to my understanding is without CoT or any other prompts that helps the LLMs to solve the problem. Then it is hard to compare "behaviours" vs existing CoT techniques when both prompt LLMs to systematically solve the given problems. In fact, the critique-and-revise baseline (line 351) is much stronger than the baseline used in BCI, and Figure 5 seems to show that baseline models prompted this way will possibly achieve fair performance. Combining Figure 5 and Figure 3/4, it seems to me that the improvement brought by BCI is marginal and could be outperformed by other prompting techniques.

2. There is no detailed analysis on the discovered "behaviours". Performance improvement is not enough to draw generalisable insights. I would like to see in depth analysis on questions such as: (1) what information are they providing the the LLMs? (2) can the paper give a categorisation of them and elaborate how such information is superior than what could be provided via other prompt techniques?

3. The generalisability of the "behaviours" are questionable. Although the paper has shown that "behaviours" can generalise across models and across different versions of AIME, it is unclear to me whether it is generalisable and scalable for other tasks beyond math problems. In fact, LLMs are being used in a wide range of tasks other than math which requires reasoning capabilities. So the current empirical results are limited in the scope.

**Questions:**

1. The empirical results lack comparisons with other prompting techniques and it is unclear to me how "behaviours" are special or better than other existing CoT methods.

2. The paper lacks analysis about the discovered behaviours. Please see the weakness point 2.

3. Figure 7 shows that some LLMs' performance on AIME-25 is significantly worse than on AIME-23/24. The performance is also inconsistent among LLMs. What is the cause for this?

4. Please refer to other points in the "weaknesses"

---

### Official Review · Reviewer_R5fR · 2025-11-02

**Soundness:** 2
**Presentation:** 2
**Contribution:** 3
**Rating:** 4
**Confidence:** 4

**Summary:**

This paper proposes a novel approach to reducing redundant reasoning tokens in LLMs to improve inference efficiency. The key idea is to precompute a collection of common reasoning behaviors, called a "reasoning handbook", using a set of in-distribution samples. These are referenced by the model at test-time to quickly re-apply the same reasoning patterns on novel samples thereby, e.g., saving tokens on re-deriving the same technique. The authors propose three main methods to apply this handbook: (1) behavior conditioned inference (BCI) which provides (part of) the handbook in-context, (2) behavior-guided self-improvement, in which the model essentially generates a reasoning handbook for each individual question at test-time, and (3) (BC-SFT) SFT on teacher-generated paths using method (1).

**Strengths:**

- The idea is interesting and sound
- The authors compare methods at various thinking budgets on thinking-enabled models to identify a output tokens to accuracy curve for each method for effective pareto analysis.
- The authors apply the key idea of using a behavior handbook to three (or four) concrete methods based on common model deployment strategies, and study these methods
- The BCI method shows significant pareto improvement on output token length to task accuracy on the MATH dataset.
- The self-improvement method achieves significant accuracy gains on AIME.

**Weaknesses:**

These are the key factors that I believe are relevant to this idea, on which I base my critique below:
- How much does the method improve pareto-efficiency between cost and task performance?
- How well does the method generalize across tasks, or how difficult is it to apply to a new task?
- How does the method manage the size of the handbook, which could be potentially very large?

I will address weaknesses for each individual method variant.

(1) Common

The following weaknesses have been discussed in the limitation section.

- **W1**. All method variants require, (1) a large dedicated training set and (2) significant inference computation to prepare the behavior handbook, **for each task**.
- **W2**. The method is only evaluated on math tasks (MATH and AIME-XX), and it is unclear whether it can be applied to other tasks. This includes uncertainty on, e.g., how well the method improves efficiency on other tasks, whether "behaviors" can be effectively identified for other tasks, etc.

(2) BCI

(2-1) MATH

As mentioned above, the BCI method shows significant pareto improvement on output token length to task accuracy on the MATH dataset. However:

- **W3**. The relevant behaviors for the MATH dataset are filtered based on the topic of the given test sample, to mitigate context length. However, this heuristic requires that each behavior is annotated with a predefined set of topics, and may not be applicable to other tasks.

(2-2) AIME (with RAG)

Note that I differentiated this method variant from that used on MATH, because the authors apply RAG to handle the large size of the handbook on the AIME dataset. While token efficiency generally improves, the improvement is not as much as that on MATH.

(2-Overall)

- **W4**. The authors apply two different variants of the BCI method on MATH and AIME tasks, respectively. Thus, evaluation of each variant is limited to one task.

W4 could be addressed by evaluating the RAG-based method on MATH, potentially demonstrating a single method that can be applied to various tasks, with consistent performance gains.

(3) Behavior-guided self-improvement

- **W5**. (Minor) while the maximum accuracy improves significantly, the efficiency (output tokens to accuracy curve) is degraded.

(4) BC-SFT

- **W6**. The authors do not compare with previous baselines such as [1], which generates an SFT dataset with concise reasoning paths by selecting shortest of N samples.

[1] Munkhbat 2025, Self-Training Elicits Concise Reasoning in Large Language Models

**Questions:**

1. How long is each behavior handbook (in BCI), in terms of tokens? While the authors address input costs on the Efficiency Considerations paragraph on page 6, the cost overhead may be still be significant if the handbook is excessively long.
2. Does the average number of tokens in Figure 5 include output tokens across all steps, including those used to derive the behavior handbook, including the 16 reasoning traces? If so, the number seems very low.

---

### Meta-Review · Area_Chair_FCBT · 2026-01-18

**Summary:**

Reviewers agree the core idea—extracting reusable “behaviors” from recurring CoT fragments and using a behavior handbook—is interesting, and one reviewer finds the experiments extensive and the approach clearly beneficial.

However, two reviewers recommend reject and raise decision-driving concerns about (i) baseline strength / attribution of gains (BCI compared against a weak baseline; improvements may be marginal relative to stronger prompting), (ii) token-efficiency accounting (output-token savings may be offset by large input prompts and quadratic attention cost), and (iii) limited scope / generalization (evaluations mostly on math; unclear scalability beyond math and how to manage large handbook size).

**Reviewer Concerns:**

I did not see a rebuttal, so no concerns were addressed via rebuttal.

**Reviewer Scores:**

There’s no rebuttal, so I assume minimal movement

---

### Decision · Program_Chairs · 2026-01-26

Reject